# Secretory IgA-ETEC F5 Immune Complexes Promote Dendritic Cell Differentiation and Prime T Cell Proliferation in the Mouse Intestine

**DOI:** 10.3390/life13091936

**Published:** 2023-09-20

**Authors:** Da Qin, Ying Li, Xiaoyan Chen, Liyang Li, Guihua Wang, Xilin Hou, Liyun Yu

**Affiliations:** 1College of Life Science and Technology, Heilongjiang Bayi Agricultural University, Daqing 163319, China; da_qin0130@126.com (D.Q.); liying7697@163.com (Y.L.); chen_xiaoyan2023@126.com (X.C.); liyang_li0313@126.com (L.L.); wgh645@163.com (G.W.); 2College of Animal Science and Technology, Heilongjiang Bayi Agricultural University, Daqing 163319, China; houxilin@byau.edu.cn

**Keywords:** SIgA–ETEC F5 immune complex, enterotoxigenic *Escherichia coli* (ETEC), F5 fimbrial protein, secretory IgA, mucosal immune response

## Abstract

Although secretory IgA (SIgA) is the dominant antibody in mucosal secretions, the capacity of the SIgA–antigen complex to prime the activation of dendritic cells (DCs) and T cells in the intestinal epithelium is not well understood. To this end, the SIgA–ETEC F5 immune complexes (ICs) were prepared via Ni-NTA pull-down. After injecting the ICs into the intestines of SPF BALB/c mice, most ICs were observed in the Peyer’s patch (PP). We established a microfold (M) cell culture model in vitro for transport experiments and the inhibition test. To evaluate the priming effect of mucosal immunity, we employed the DC2.4 stimulation test, T lymphocyte proliferation assays, and cytokine detection assays. We found that the ICs were taken up via clathrin-dependent endocytosis through M cells. The high expression of costimulatory molecules CD86, CD80, and CD40 indicated that the ICs promoted the differentiation and maturation of DC2.4 cells. The stimulation index (SI) in the complex group was significantly higher than in the control group, suggesting that the ICs stimulated the proliferation of primed T cells. The secretion of some cytokines, namely TNF-α, IFN-γ, IL-2, IL-4, IL-5, and IL-6, in spleen cells from the immunized mice was upregulated. These results indicate that ETEC F5 delivery mediated by SIgA in PPs initiates mucosal immune responses.

## 1. Introduction

As the most abundant type of antibody in the animal intestine, SIgA is assumed to be the first line of defense to protect the body from intestinal pathogenic microorganisms and toxins [1]. Specific SIgA in the mucosa has been recognized as the standard for evaluating mucosal immunity [2]. Peyer’s patches were covered with follicle-associated epithelium (FAE), containing microfold cells (M cells) inside. The underlying DCs can capture mucosal antigens by extending dendrites or through transcytosis by M cells. There is also evidence that both FAE and small intestinal goblet cells (GCs) are involved in antigen uptake [3,4,5,6]. An intriguing feature of SIgA in the enteric canal is its capacity to cross the intestinal barrier via M cells overlying Peyer’s patches and to subsequently associate with underlying DCs in the subepithelial dome region, either in the form of free antibodies or in complex with antigens [7,8,9,10].

Immune complexes are formed as a result of noncovalent interactions between foreign antigens or autoantigens and antibody proteins [11]. In both the poultry industry and clinical healthcare settings, ICs have been used as preventive and therapeutic vaccines [12,13,14,15]. However, most of these studies focus on the IgG immune complex; few studies consider the specific SIgA–antigen complex. Enterotoxigenic *Escherichia coli* (ETEC) F5 is the common etiologic agent among the six recognized diarrheagenic categories of *E. coli* [16]. ETEC F5 is a pathogen causing severe diarrheal disease in suckling and weaning animals manifested by a frequently fatal secretory diarrhea in a clinical setting [17]. In pig-specific ETEC strains, there are five fimbriae: F4 (K88), F5 (K99), F6 (987P), F41, and F18 [13,18,19,20,21,22].

In this study, we prepared SIgA–ETEC F5 immune complexes and found a higher number of these complexes in PPs after their injection into the small intestine loop in mice. To test the uptake efficacy and transport mechanism of the immune complex, there is a need for an in vitro M cell model. The inhibition test of the M cell uptake mechanism was carried out with this model via the ingestion of the immune complexes by M cells. Furthermore, we investigated whether the SIgA–ETEC F5 immune complex promoted the differentiation and maturation of DC2.4 cells and stimulated the proliferation of primed T cells. DC 2.4 is a cell line comprising dendritic cells, which are commonly used to mimic in vivo dendritic cell stimulation. Oral administration is the means through which the local immune response is induced; however, it is unclear whether the SIgA–ETEC F5 immune complexes effectively prime the local mucosal immune response. To solve this problem, we analyzed the expression levels of costimulatory molecules CD86, CD80, and CD40 on the surface of DC2.4 cells stimulated by the ICs. We also investigated the secretion of TNF-α, IFN-γ, IL-2, IL-4, IL-6, and IL-5 cytokines in the spleen cells from the immunized mice and T lymphocyte proliferation response.

## 2. Materials and Methods

### 2.1. Bacterial Strains and Growth Conditions

The pET-32a expression vector, competent cell *E. coli* BL21 (DE3), and virulent ETEC F5 (C83912) were all preserved by the Genetic Engineering Laboratory, College of Life Science and Technology, Heilongjiang Bayi Agricultural University. The *E. coli* strains were cultured at 37 °C with continuous shaking in a Luria–Bertani (LB) medium.

### 2.2. Expression and Purification of the F5 Fusion Protein

The 498 bp DNA fragment encoding the F5 fimbrial protein was amplified using PCR with primers 5′-CGCGGATCCAATACAGGTACTATTAAC- 3′ and 5′-CGCAAGC TTCATATAAGTGACT-3′. The PCR product was inserted into vector pET-32a to construct the recombinant plasmid pET-32a-F5 after digestion with *Bam*H I and *Hind* Ⅲ (Takara, Tokyo, Japan). PCR amplification was performed as follows: 94 °C for 5 min; 35 cycles of 94 °C for 30 s, 56 °C for 30 s, and 72 °C for 45 s; and 72 °C for 10 min for the final cycle.

The his-tagged F5 protein was expressed in *E. coli* BL21 for 6 h at 37 °C in an LB medium after being induced by 1.0 mM of isopropyl-β-d-thiogalactoside (IPTG). The recombinant protein was purified with Ni-NTA agarose (QIAGEN, Duesseldorf, Germany), which was verified using Western blotting with the anti-F5 antibody as the primary antibody.

### 2.3. Preparation of Specific SIgA

Female C57BL/6 mice (6–8 weeks old), which were purchased from Hongda Animal Laboratories, Changchun, China, were used in all experiments. All of the food and water were autoclaved according to the animal protocols approved by the Institutional Animal Care and Use Committee (IACUC). Mice were acclimated to the new environment for 1 week before being used for immunization. The use of animals for this experiment was approved by the Heilongjiang Bayi Agricultural University Institutional Animal Care and Use Committee.

The SPF BALB/c mice in the experimental group were fed intragastrically via 1 × 10^11^ CFU/mL ETEC F5 (C83912) in a 200 μL bacterial suspension per mouse, while the mice in the control group were fed with 200 μL PBS per mouse. The mice in the blank group were not treated. Feces samples of 0.1 g each were collected on Days 0–4, 20–24, and 39–43, and then dissolved in 1 mL of PBS by shaking in a horizontal shaker at 4 °C. The supernatant containing specific SIgA antibodies was collected via centrifugation. The SIgA antibodies were concentrated using the ammonium sulfate precipitation method.

### 2.4. Pull-Down Assay for SIgA–ETEC F5 Complexes

The pull-down assay was performed using the Ni-NTA agarose immunoprecipitation kit (QIAGEN, Duesseldorf, Germany). First, the F5 protein was bound to Ni-NTA agarose overnight on a horizontal shaker at 4 °C. The agarose was rinsed three times with 0.02 M imidazole before the supernatant containing anti-F5 SIgA was added. The agarose binding the SIgA–ETEC F5 was continually incubated on a horizontal shaker at 4 °C for 4 h, and rinsed again with 0.02 M imidazole three times. Finally, the specific SIgA–ETEC F5 complexes were eluted by 0.3 M imidazole, which were then identified by immunoblotting.

### 2.5. Immunoblotting Analysis

To verify the ICs, the protein extract was analyzed using 10% SDS-PAGE, which was electrotransferred to a PVDF membrane. After blocking overnight with PBST containing 4% skim milk, the membranes were incubated using anti-F5 mouse serum (1:150) and anti-his monoclonal antibody (1:1000) (Abcam, Cambridge, UK) as a primary antibody. Horseradish peroxidase (HRP) conjugated goat anti-mouse IgG (1:10,000, Immunoway, San Diego, CA, USA) was used as the secondary antibody. After washing with PBST, the membrane was treated with a hypersensitive luminescent solution for observation. For the SIgA–ETEC F5 analysis, a mixture containing 80 μL of the elution products and a 20 μL 5 × loading buffer (containing 10% SDS and 5% β-mercaptoethanol) was placed in a boiling water bath for 10 min to ensure complete dissociation of the complex. The SDS-PAGE was performed and the proteins in gels were transferred to the PVDF membrane which was incubated using both antibodies of anti-F5 mouse serum (1:150) and goat anti-mouse IgA alpha chain (1:1000, ab97231, Abcam, Cambridge, UK) for 2 h at 37 °C. The membrane was washed with PBS three times and then incubated with goat anti-mouse IgG antibodies labelled with HRP and rabbit anti-goat-HRP at 37 °C for 1 h.

### 2.6. Small Intestine Loop Ligation Experiment

After being fasted for 24 h, the mice were anesthetized by ether. The ileum with the Peyer’s patch was extracted and ligated to the loop. Then, 250 μL of the SIgA–ETEC F5 complex (2 mg/mL) was injected to the intestine loop for 6 h. The ileum loop was fixed by 4% paraformaldehyde, embedded with the OCT embedding agent, and frozen on the rapid-freezing shelf of the microtome for 30 min at −20 °C. The sections were sliced at a thickness of 5 μm. The sliced Peyer’s patch and small intestine tissue were spread on the slides and then stained.

The tissue sections were fixed using ice-cold acetone for 30 min. The sections were washed three times with PBS before blocking with PBS containing 5% BSA for 1 h. The goat anti-mouse IgA alpha chain antibodies (1:1000, ab97231, Abcam, Cambridge, UK) and anti-his monoclonal antibody (mAb,1:1000) were added to the slices, which were incubated at 37 °C for 2 h in an immunohistochemical humidified box containing PBS. The sections were washed three times with PBS, and then doped with the FITC rabbit anti-goat IgG (H+L) antibodies (1:50, AS024, ABclonal, Wuhan, China) and Cy3 goat anti-mouse IgG (H+L) (1:50, AS008, ABclonal, Wuhan, China) as the secondary antibodies. The treated sections were incubated in the dark at 37 °C for 1 h. After washing with PBS three times, the sections were dried, and the DAPI working solution (0.1%) was added at room temperature for 10–20 min. Finally, a 5 μL anti-light attenuation mounting solution was added dropwise to the sections. The images were viewed under a confocal microscope (Leica, Jena, Germany).

### 2.7. Immunofluorescence Microscopy and Flow Cytometry

To detect the expression levels of costimulatory molecules CD86, CD80, and CD40 on the surface of DC2.4 cells in each group, flow cytometry and immunohisto-chemistry experiments were employed. The DC2.4 cells were cultivated in 12-well plates (Corning, San Diego, CA, USA). The wells were treated with different stimuli, such as the SIgA–ETEC F5 complex (500 μg), the F5 antigen (500 μg), PBS, and the 1640 medium, and continually cultured with 5% CO_2_ for 48 h at 37 °C. After washing with PBS, FITC anti-mouse CD86 antibodies (500 μL), FITC anti-mouse CD80 antibodies (500 μL), and FITC anti-mouse CD40 antibodies (500 μL) (Abcam, Cambridge, UK) were added to the wells, which were continually cultured at 37 °C for 30 min. The results were observed via immunofluorescence microscopy (NiKon, Tokyo, Japan).

The DC2.4 cells cultivated in 4.5 cm^2^ plates were treated with the abovementioned different stimuli for 48 h at 37 °C. The cells were scraped down and washed with PBS. The cells were resuspended in a 500 µL binding buffer containing 5% BSA; then, FITC anti-mouse CD86 antibodies (500 μL), FITC anti-mouse CD80 antibodies (500 μL), and FITC anti-mouse CD40 antibodies (500 μL) were added, before being incubated for 30 min at 37 °C in the dark. The cells were immediately analyzed using flow cytometry (CytoFLEX, Beckman Coulter, San Diego, CA, USA).

### 2.8. Establishment of an M Cell Culture Model In Vitro

The M cell culture model was developed following the protocol of Anne des Rieux [22]. Caco-2 and Raji cell lines were preserved by the Genetic Engineering Laboratory of College of Life Science and Technology, Heilongjiang Bayi Agricultural University. Briefly, the Caco-2 cells (5 × 10^5^) suspended in DMEM containing 10% FBS (fetal bovine serum, Gibco, CA, USA). The 1% (*v*/*v*) PEST (penicillin–streptomycin solution, Gibco CA, USA) was added on the upper insert side of the transwell (Corning, CA, USA), where Caco-2 cells were seeded. The Caco-2 cells were cultivated with 5% CO_2_ at 37 °C for 3–5 days. The inserts were inverted in a Petri dish filled with DMEM supplemented with 10% FBS + 1% (*v*/*v*) PEST. A piece of a silicon tube was then placed on the basolateral side of each insert. The cells were cultivated for another 9–11 days. The medium in the basolateral compartment was changed every other day. The Raji cells (2.5 × 10^4^) resuspended in RPMI-1640 supplemented with 10% FBS containing 1% (*v*/*v*) PEST were then added to the basolateral compartment of the inserts. The co-cultures were maintained for 5 days. The Caco-2 cells without the Raji cells were used as controls. Before the experiments, the silicon tubes were removed, and the cell mono-layers in multi-well plates were washed twice with HBSS. The inserts were used in their original orientation for all the subsequent experiments.

### 2.9. Identification of M Cells in the Culture Model In Vitro

The M cells were detached from the membranes on the transwell by the enzymolysis approach. We added anti-Gp2 polyclonal antibodies (1:200) to the cells under shaking at 37 °C for 30 min. The cells were washed 3 times with PBS. Additionally, the cells with added goat anti-rabbit FITC antibody (1:200) were incubated at 37 °C for 30 min. The cells were tested by flow cytometry after washing with PBS 3 times.

To identify the transportation function of M cells in the model, we employed PLGA microspheres as indicator agents for the transfer experiment. Then, 15 μL of green fluorescent PLGA microspheres (diameter 1 μm, LGFG1000, Sigma, San Diego, CA, USA) was mixed with a 1 mL HBSS solution in a warm bath at 37 °C up to 10^7^ particles/mL. After removing 500 μL of the mix from the transwell upper chamber containing M cells, then adding 1 mL HBSS to the lower chamber, the transwell was incubated for 6 h at 37 °C. The samples at 3 h and 6 h were taken. The number of fluorescent ions was detected per unit of time (1 min) through flow cytometry. The HBSS was as negative control in the same condition.

### 2.10. The Uptake Mechanism of the SIgA–ETEC F5 Complex by M Cells in the Culture Model

The SIgA–ETEC F5 complexes (1 mg in 500 μL elution buffer) were added to a transwell upper chamber containing M cells, with a 1 mL HBSS in the lower chamber. The liquid in the lower chamber was sampled after being taken up for 3 h and 6 h at 37 °C. The liquid samples were analyzed by Western blot to identify the successfully transported SIgA–ETEC F5 complexes.

For inhibition tests of macropinocytosis and non-clathrin-mediated endocytosis, 500 μL M cells and a 1 mL inhibitor, EIPA 10 μM or/and mycostatin 50 μg/mL (absin, Beijing, China), were added to the upper and low chambers, respectively. The transwells were incubated at 37 °C for 20 min. Then, 500 μL of SIgA–ETEC F5 complex was added to M cells in the transwell upper chamber and continually incubated for 6 h. Liquid samples were taken for detection of the SIgA–ETEC F5 complex by Western blot.

For the inhibition of clathrin-mediated endocytosis, first, the transepithelial resistance was measured. Second, a potassium depletion test was performed. The M cell monolayer was subjected to hypotonic shock in DMEM/H_2_O (1:1) at 37 °C for 5 min. The cells were washed once with potassium-free HBSS and incubated in the medium at 37 °C for 1 h; the transepithelial resistance was then measured again. A 500 μL sample of the SIgA–ETEC F5 complex was added to the upper transwell chamber containing M cells. Samples were taken after incubation at 37 °C for 3 h and 6 h, and liquid from the lower chamber was taken for identification of the SIgA–ETEC F5 complex by Western blot.

### 2.11. Immunological Assays

For the T cell proliferation assay, the cell counting kit-8 assay (CCK-8) (ABclonal, Wuhan, China) was carried out. Twelve mice from each group on Day 57 post infection with 1 × 10^11^ ETEC F5 (C83912) were killed to make a single-cell spleen suspension using EZ-SepTM Mouse 1 × Lymphocytes Separation Medium (DaKeWe Biotech Company Limited, Shenzhen, China), according to the manufacturer’s instructions. The splenocytes were adjusted to a concentration of 10^6^ cells/mL in the RPMI 1640 complete medium. The cells were seeded into a 96-well microtiter plate (standard flat-bottomed wells, Corning) with 100 μL per well. Then, stimulators of 5 μg/mL ConA (Sigma; positive control group), 500 μg SIgA–ETEC F5 complex, 500 μg F5 antigen, or PBS (background group), were added into each well, with three duplications of each group. The plate was incubated at 37 °C for 24 h in a humidified atmosphere containing 5% CO_2_. To assess cell proliferation, the plate was incubated with 10 μL CCK-8 per well for 2 h. The plate was read at 450 nm using an autoreader (Molecular Devices), and the stimulation index (SI) was calculated as A450 (treatment group)/A450 (negative control).

To test whether the specific SIgA–ETEC F5 complex primed an immune response, single-cell suspensions of splenocytes isolated from the immunized mice were forced through a 200 μm nylon cell strainer and cultivated at 37 °C for 24 h. The cytokines of interferon-γ, interleukin-4, interleukin-5, tumor necrosis factor-α, interleukin-6, and interleukin-2 were quantified by the enzyme-linked immunospot (ELISPOT) assay (DaKeWe Biotech Company Limited, Shenzhen, China). Briefly, the plates were incubated with an RPMI-1640 complete medium for 10 min; then, 100 μL of splenocytes (10^5^ cells per well) was added to each well. The stimuli, 5 μg/mL ConA (Sigma; positive control group), the SIgA–ETEC F5 complex (500 μg), the F5 antigen (500 μg), or PBS (background group) were added into each well, with three duplications in each group. Following incubation as described by the kit manufacturer, the treated plates were dried and sealed, and then the number of spots was recorded for statistical analysis.

### 2.12. Statistical Analysis

Data are presented as the mean ± standard deviation. Statistical analyses were performed using GraphPad Prism 7.0 software. Differences between groups were evaluated by one-way analysis of variance (ANOVA) followed by the LSD, and *p* < 0.05 was considered statistically significant.

## 3. Results

### 3.1. Identification of the Purified His-F5 Protein and Specific SIgA–ETEC F5 Complex

In order to prepare SIgA–ETEC F5 ICs, we first expressed and purified the his-tagged F5 fusion protein in recombinant *E. coli*. After performing analysis with the immunoblotting technique, the his-tagged F5 protein and the specific SIgA–ETEC F5 complex were identified (Figure 1). The sizes of the F5 protein and the disaggregated F5 from ICs were approximately 37 KDa with mouse anti-F5 serum as the primary antibody (Figure 1A). The SIgA–ETEC F5 ICs were dissociated by SDS and β-mercaptoethanol. The disaggregated SIgA was 56 KDa in the presence of the mouse IgA alpha-chain antibody (Figure 1B). Both bands were observed when the blotting membrane was co-incubated with the two above antibodies (Figure 1C). However, no clear band was observed in pET-32a/*E. coli* as a control (Figure 1C). The results showed that the specific SIgA–ETEC F5 immune complex was successfully prepared.

### 3.2. Results of the Small Intestine Loop Ligation Experiment

To detect where the SIgA–ETEC F5 ICs were taken up in the gut, small intestine loop ligation experiments were performed. Immunohistochemistry results of the mouse ileum intestine loop ligation showed that the SIgA–ETEC F5 complex was taken up through the Peyer’s patches. As shown in Figure 2, in the Peyer’s patches (A1–D1), the yellow (green + red) could be clearly seen (D1), but not in the small intestine tissue without PPs (A2–D2). The intestinal tissue was blue, and SIgA in the complex was green. The F5 was stained with PE in red. When merging the two pictures, red encountered blue and changed to yellow, indicating that F5 and SIgA were in a protein structure. These results suggested that SIgA-antigen ICs, similarly to other antigens in the gut, were mainly up taken from PPs.

### 3.3. The SIgA–ETEC F5 Complex Promoted the Differentiation and Maturation of DC2.4 Cells

To mimic in vivo dendritic cell stimulation, we used SIgA–ETEC F5 ICs to stimulate DC2.4 cells. The expression levels of costimulatory molecules CD86, CD80, and CD40 in DC2.4 cells were detected through immunofluorescence microscopy and flow cytometry (Figure 3). The cells in all groups were viewed under white light (Figure 3A–D). Under fluorescent light, positive cells were sorted out and observed through microscopy (Figure 3A–C). However, green fluorescence was not detected in the control group (Figure 3D). Additionally, the merged figures showed the same results in Figure 3A–D. FACS was used to further analyze the expression of the costimulatory molecules CD86, CD80, and CD40 (Figure 3E–H). The results in the SIgA–ETEC F5 complex group showed higher green fluorescence intensity than those in the F5 and control groups (Figure 3E–G). The cells treated with PBS were used as controls. Importantly, the SIgA–ETEC F5 complex exhibited a significantly higher fluorescence intensity than the F5 antigen (Figure 3H). These results indicated that the SIgA–ETEC F5 complex strongly promoted the differentiation and maturation of DC2.4 cells.

### 3.4. Identification of the M Cell Culture Model In Vitro

To determine the efficient uptake of SIgA–ETEC F5 ICs by M cells in the small intestine PP, we developed an in vitro model for M cells. The expression level of Gp2 protein on the surface of M cells was detected using FACS to prove the maturation of M cells (Figure 4A,B). The conversion rate of M cells was 36.4% (Figure 4B); however, in Caco-2 cells, it was only 1.23% (Figure 4A). The M cells transported PLGA microspheres, and the numbers increased over time (Figure 4D–F). Caco-2 cells did not transport the microspheres (Figure 4C). The results show that the culture model of M cells in vitro was successfully established.

### 3.5. Ingestion of the SIgA–ETEC F5 Complex by M Cells Mainly Relying on Clathrin-Dependent Endocytosis

The mechanism for SIgA–ETEC F5 complex transportation via M cell was detected by Western blot (Figure 5).

When we inhibited macropinocytosis, clathrin-independent endocytosis, micropinocytosis, clathrin-independent endocytosis (non-clathrin-mediated), and clathrin-mediated endocytosis, the amounts of SIgA–ETEC F5 complex ingested by M cells all decreased compared with the unprocessed control (Lane 1–5), regardless of whether they were processed for 3 h or 6 h. However, the largest decline was observed under the suppression of clathrin-dependent endocytosis (Lane 5). The results also showed that the degree of uptake and transport increased with the treatment time, but Caco-2 cells did not transport the complexes (Lane 6). These results indicated that clathrin-dependent endocytosis of the M cells was the largest influencing factor of complex intake.

### 3.6. Characterization of the Immune Responses in BALB/c Mice

To further analyze whether the SIgA–ETEC F5 immune complex could induce a cell-mediated immune response, the T lymphocyte proliferation response was tested with CCK-8 (Figure 6). As shown in Figure 6A, the evaluation of T cell proliferation indicated that the stimulation index (SI) of mice splenic T lymphocytes in each group was significantly increased. Importantly, the SI of the specific SIgA–ETEC F5 immune complex was significantly higher than that of the free F5 antigen and the control groups.

The ELISPOT assay was employed to identify the cytokines secreted in splenic lymphocytes stimulated by SIgA–ETEC F5 immune complexes (Figure 6B–G). As shown in Figure 6B–G, the secretion levels of IFN-γ, IL-2, TNF-α, IL-4, IL-5, and IL-6 were all significantly increased in the SIgA–ETEC F5 immune complex group and the free F5 antigen group compared with the PBS group. Although the SI of the ICs was not as high as ConA (positive control), it was significantly different from that of PBS (*p* < 0.001). More importantly, the levels in the ICs group were also significantly different from those in the free F5 antigen group (*p* < 0.001).

## 4. Discussion

Previous studies have demonstrated that SIgA directs bacterial cargo to the subepithelial dome (SED) region, where DC and M cells sample the SIgA-Ag immune complex [13]. To examine whether SIgA Ab can deliver an associated bacterial Ag to PPs, SIgA–ETEC F5 ICs were prepared. In our study, the preparation of a large number of specific SIgA monoclonal antibodies was very difficult, with high technical complications involved in the preparation of the ETEC F5-SIgA immune complex. In order to solve this problem, the prokaryotic expression vector of pET-32a-F5 was constructed, including the expressed recombinant protein as the inclusion body. Thus, the purification of recombinant F5-his fusion protein was carried out under denaturation conditions. A large number of purified recombinant proteins were renatured by removing urea through dialysis. SIgA, the key antibody in mucosal immunity, mainly exists in gastrointestinal juice, milk and other exocrine fluids, as well as feces. To obtain the anti-F5-specific SIgA antibody in feces, the mice were intragastrically administrated with ETEC F5, and then boosted twice. The collected feces were dissolved in PBS to prepare fecal supernatants. To enrich antibodies, we used the ammonium sulfate precipitation method to concentrate the supernatant containing specific SIgA antibodies. The immune complex was prepared efficiently using the his-F5 pull-down method. The SIgA–ETEC F5 immune complexes were eluted up to the milligram level, which were dissociated by a reductant and identified by Western blot. After exploring the experimental conditions numerous times, the preparation method of the immune complex was established, indicating that we overcame this technical bottleneck in this study.

Peyer’s patches play a central role in the induction of mucosal immune responses; full-time antigen presenting cells (APCs) play an indispensable role [23]. Rapid antigen recognition and presentation produce a fast and effective immune response [24]. As key to initiating the mucosal immune response, M cells exhibit a variety of antigen uptake mechanisms to achieve this effect [4,25]. For DCs as the most important antigen presenting cells, their presentation ability is also very important in mucosal immunity [26]. M cells internalize and transport large granular material, such as bacterial ICs. There are several internalization mechanisms: macropinocytosis, grid-independent endocytosis, and grid-dependent endocytosis. In this study, we found that these three internalization mechanisms were all affected by the endocytosis of M cells, in which grid-dependent endocytosis had the greatest influencing factor, because grid protein transported more ICs than the other two mechanisms combined. Given that the uptake of the SIgA–ETEC F5 complex by M cells was mainly mediated by grid protein, it is fully conceivable that immune complexes can be absorbed by intestinal M cells and initiate mucosal immune responses in vivo. After the DC2.4 cells were stimulated by immune complexes, we found that immune complexes could promote the maturation of DCs more than free antigen stimulation, indicating that immune complexes enhanced the ability of antigen processing and presentation to lymphocytes.

One of the important characteristics of the mucosal immune response is the production of SIgA [23], which plays a key role in resisting the adhesion and invasion of intestinal pathogens, as well as the neutralization of pathogens [27]. Controlling the synthesis and secretion of SIgA and the immune response mediated by SIgA depends on the helper T cells (Th) [28] that regulate immunity by secreting cytokines. Th1 cells secrete high levels of IFN-γ, IL-2, and TNF-α, while Th2 cells mainly secrete IL-4, IL-5, and IL-6. It has been reported that IL-4 helps to induce the transformation of B cells into IgA^+^-secreting cells [29]. IL-4 also cooperates with other cytokines such as IL-5 and IL-6 to stimulate the growth and differentiation of B cells, enhancing the ability of mature B cells to synthesize IgA, and thus promoting the production of IgA. IFN-γ increases the secretion and transport of IgA [30]. In this study, it was found that SIgA–ETEC F5 ICs increased the secretion of lymphocyte cytokines, more significantly than that of free antigen alone, suggesting that the ICs enhance the synthesis and secretion of SIgA by upregulating these cytokines, and can maintain health by exercising the function of SIgA antibodies.

T lymphocytes are the most important immune helper cells in the immune response, and IL-2 is the main cytokine that causes T cell proliferation. TNF-α can enhance T cells to produce cytokines dominated by IL-2, and they interact with each other to promote T cell proliferation. In this study, we proved that the SIgA–ETEC F5 immune complex upregulated the secretion of IL-2 and TNF-α more effectively than free antigen stimulation. The T lymphocyte proliferation test also confirmed that SIgA–ETEC F5 was able to enhance the activation, proliferation, and differentiation of T lymphocytes, thus producing more cytokines, and further promoting the secretion of SIgA.

The evaluation of the mucosal immune protective effect for candidate IC vaccines still needs to be verified through challenge experiments. If the prepared SIgA–ETEC F5 ICs were to be directly eaten, they would be degraded by pepsin and lose their effects [31]. Therefore, we will continue to explore effective oral antigen delivery vectors, such as polymer particles (MPs) and nanoparticles (NPs). Polymer particles (MPs) and nanoparticles (NPs) have been widely used in the development of subunit vaccines. Immune complexes can be wrapped in polymer particles to form microcapsules in order for them to smoothly pass through the gastrointestinal tract and mucous layer, as well as to slowly release the antigen–antibody complex in the intestinal tract. Therefore, in the follow-up of this study, we will further design candidate ICs vaccine in these aspects for oral immunization.

In conclusion, the F5 antigen reverse transportation mediated by SIgA to the intestinal PPs served to initiate the mucosal immune responses in this study. It was found that the SIgA–ETEC F5 ICs were mainly absorbed and transported by M cells through endocytosis, mainly mediated by grid protein. When the complex encountered DCs in the cystic structure of M cells, on the one hand, it promoted the maturation of DCs; on the other hand, it was regulated and swallowed by DCs, thus priming the intestinal mucosal immune response. We also observed that the SIgA–ETEC F5 immune complex promoted the expression and secretion of related cytokines, as well as the activation and proliferation of T lymphocytes. However, evaluation of the mucosal immune protection effect for SIgA–ETEC F5 immune complexes still needs animal challenge tests to determine oral immunity in vivo. There are many more questions regarding SIgA–antigen ICs, such as the delivery method of immune complexes, more efficient SIgA purification methods, and the effect of its large-scale application as a vaccine.

## Figures and Tables

**Figure 1 life-13-01936-f001:**
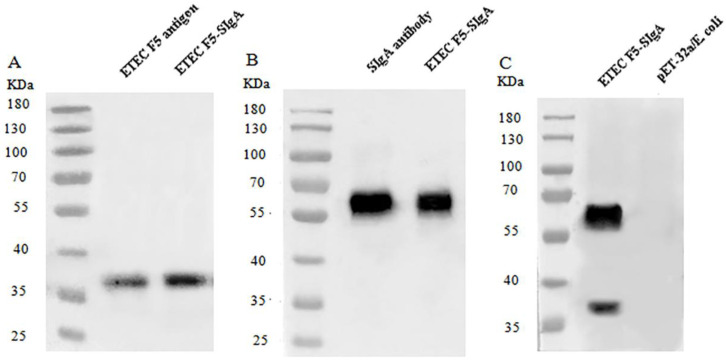
Identification of the disaggregated SIgA–ETEC F5 by immunoblotting, with anti-F5 mouse serum as the primary antibody (**A**); mouse IgA alpha-chain mAb as the primary antibody (**B**); and both anti-F5 mouse serum and IgA alpha-chain mAb (**C**).

**Figure 2 life-13-01936-f002:**
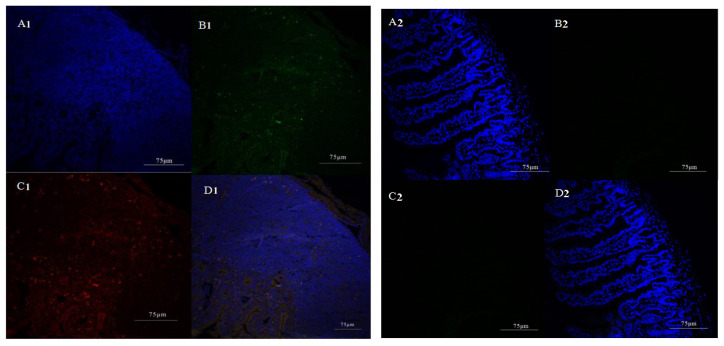
Immunohistochemistry of Peyer’s patches (**A1**–**D1**) and ileum tissue (no PPs, (**A2**–**D2**)) after ligating the mouse intestine loop. After the loop was treated with the SIgA–ETEC F5 complex for 6 h, fluorescence was observed in sections. The intestinal tissue stained with DAPI in blue (**A**); SIgA in the complex stained with FITC in green (**B**); F5 was stained with PE in red (**C**); and merged images (**D**). Green: SIgA; red: F5; blue: DAPI. Scale bar: 75 µm.

**Figure 3 life-13-01936-f003:**
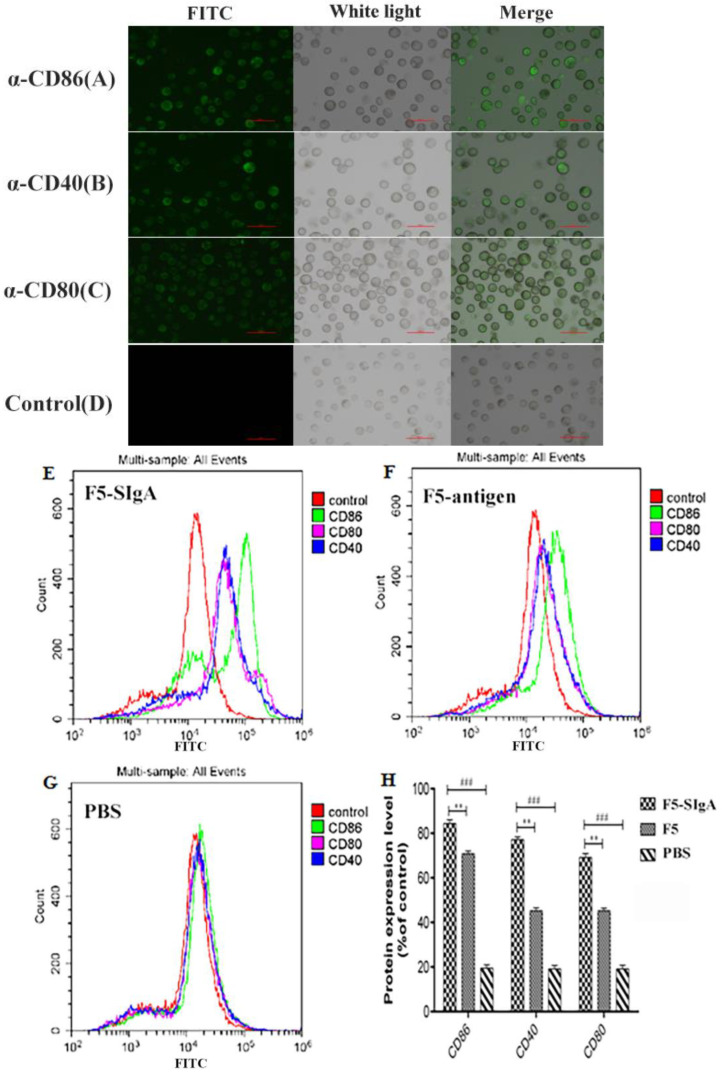
After stimulation, the expression levels of costimulatory molecules CD86, CD80, and CD40 in DC2.4 cells in each group were detected by immunofluorescence microscopy and flow cytometry. The SIgA–ETEC F5 complex stimulated the expression of CD86 (**A**), CD40 (**B**), and CD80 (**C**), but the negative control did not show green fluorescence (**D**). SIgA–ETEC F5 complex stimulation (**E**); F5 antigen stimulation (**F**); PBS stimulation (**G**). Quantification of these three costimulatory molecules by FACS (**H**). The data represent the mean ± standard deviation (SD; vertical bars) of triplicate experiments: **, *p* < 0.01; ###, *p* < 0.001.

**Figure 4 life-13-01936-f004:**
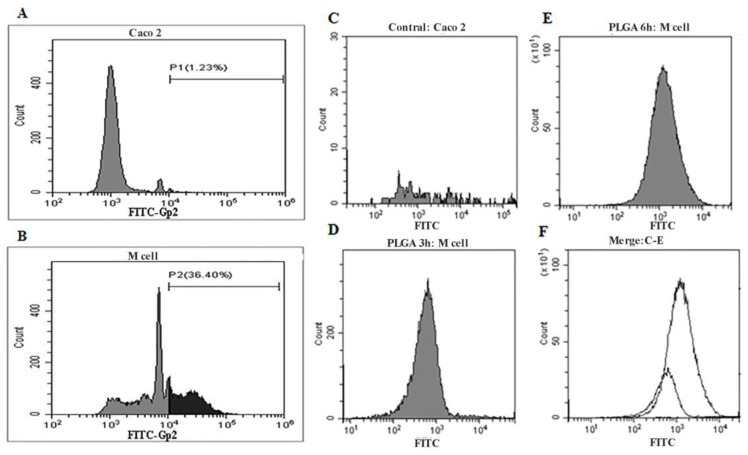
Characteristics of the M cell culture model in vitro. The expression levels of Gp2 protein on the surface of M cells. Caco-2 cells as control (**A**). M cell culture model (**B**). The M cell transported PLGA microspheres particles, as detected through flow cytometry. Caco-2 as the negative control (**C**). Incubation of the PLGA microspheres for 3 h (**D**), 6 h (**E**), and merged with contral, PLGA 6 h, PLGA 3 h (**F**).

**Figure 5 life-13-01936-f005:**
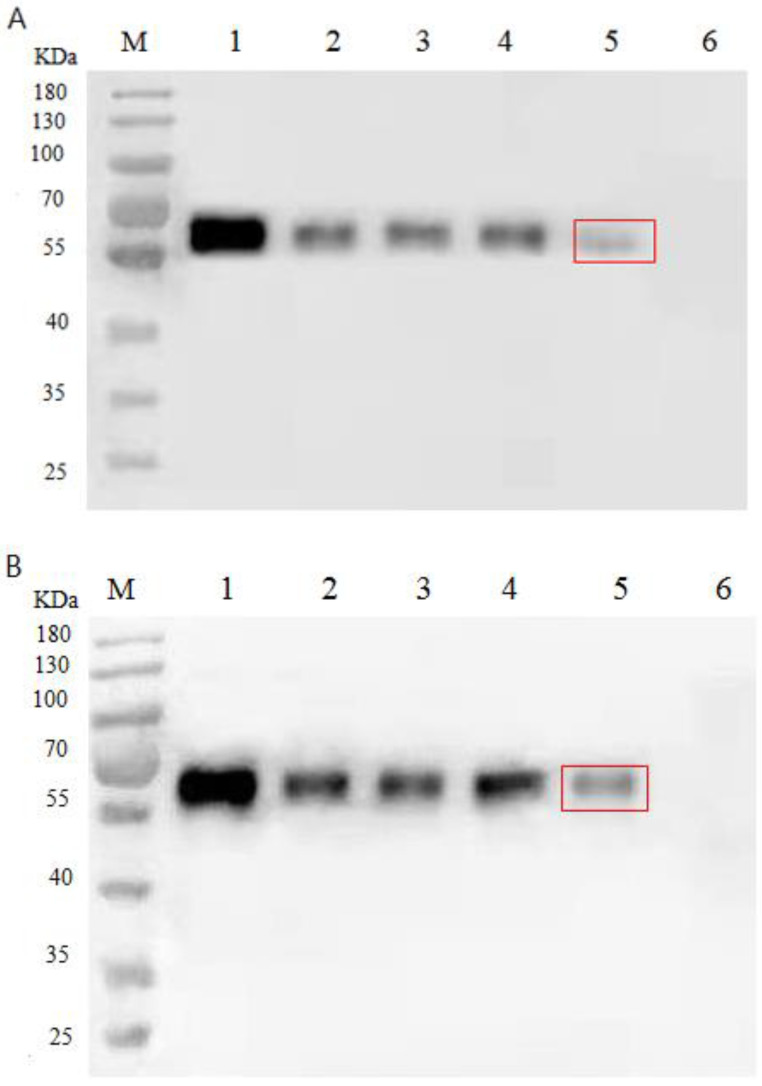
Identification of the mechanism transporting the SIgA–ETEC F5 immune complex by Western Blot. The SIgA–ETEC F5 immune complexes were transported by M cells in different treatment groups after 3 h (**A**) and 6 h (**B**). 1. Normal transport; 2. Inhibition of macropinocytosis; 3. Inhibition of clathrin-independent endocytosis; 4. Inhibition of macropinocytosis and clathrin-independent endocytosis; 5. Inhibition of clathrin-mediated endocytosis; and 6. Caco-2 transport.

**Figure 6 life-13-01936-f006:**
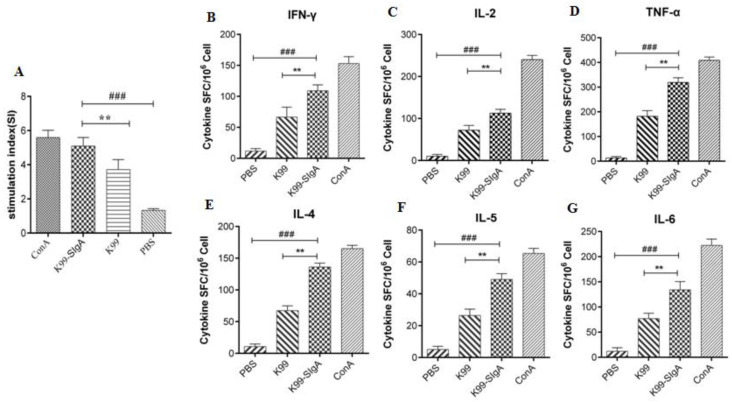
Characterization of the immune responses in BALB/c mice. T lymphocyte proliferation by CCK-8 (**A**). Levels of the cytokines IFN-γ (**B**), IL-2 (**C**), TNF-α (**D**), IL-4 (**E**), IL-5 (**F**), and IL-6 (**G**) were detected using the cytokine ELISPOT kit. Triplicate experiments: **, *p* < 0.01; ###, *p* < 0.001.

## Data Availability

No new data were created.

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
