# Peer review of "Secretory IgA-ETEC F5 Immune Complexes Promote Dendritic Cell Differentiation and Prime T Cell Proliferation in the Mouse Intestine"

_life, 2023, doi:10.3390/life13091936_

Round 1

Reviewer 1 Report

General Comments:

The paper investigates the role of the secretory IgA (SIgA)-F5 immune complex in priming the activation of DC and T cells in the intestinal epithelium. The study is well-structured and provides valuable insights into the mucosal immune responses initiated by ETEC F5 delivery mediated by SIgA in the PPs. The topic is of significant importance, given the role of SIgA in mucosal immunity. The methods employed, such as the M cell culture model, DC2.4 cells stimulation test, and cytokines detection, are appropriate for the research question. The results are presented clearly, with evidence supporting the main claims of the paper.

Areas for Improvement:

Clarity: Some sentences are lengthy and could be simplified for better understanding.

Terminology: There are instances where the terminology might be confusing for readers not familiar with the subject. For example, abbreviations like "PPs", "DC", and "ICs" should be defined upon first use in both abstract and main text.

Results: The results section is detailed. However, it might be beneficial to provide more context or interpretation alongside the results to guide the reader.

Discussion: The discussion section could benefit from a more in-depth comparison with existing literature. While the paper cites previous studies, a more detailed discussion on how this study advances the field would be beneficial.

Comments on the Results part:

3.1: Please introduce at the beginning of the results part: what is F5-SigA, and explain the rationale of this design.

3.2: It would be useful to mention what is green and red, not just in the Figure Legend but also in the text of Results. It could help the readers read easily.

3.3: Please introduce what is DC2.4, is it a cell line of dendritic cells? Please also explain at the beginning of the paragraph why to use this DC2.4 model, for example, you could mention if is it commonly used to mimic in vivo dendritic cell stimulation.

Figure 3: Instead of writing 1,2,3, please label the pictures with channel names, e.g., FITC, merge...Similarly, please label the use of antibodies on the left of the panels, e.g., α-CD80, α-CD86, α-CD40.

Please specify in the text what control was used, e.g., unstimulated DC2.4 or cells treated with PBS. This could help readers read easily.

If possible, please also briefly (in one or two sentences) write the rationale and design of the experiments at the beginning of each part of the Results. These can help increase readability and guide the readers to understand the logic of the paper.

3.4: Are microspheres and milk spheres the same thing? Please specify.

Figure 4: Please label the X-axis with the protein name, e.g., FITC-Gp2. Remove the blue and red buttons on the right of the panels. Pay attention to the labels on the top of each panel, they are hard to read and understand, please write them in the same format.

Figure 4F: There seems no negative peak but there’s a negative label. Please check carefully and correct it.

Author Response

Areas for Improvement:

Clarity: Some sentences are lengthy and could be simplified for better understanding.

 line165 “The 1% (v/v) PEST(Penicillin-Streptomycin Solution, Gibco CA, United States) were added on the upper insert side of the transwell (Corning, United States), where Caco-2 cells were seeded.

Line 332  “The ELISPOT assay was employed to detect the secreted cytokines in splenic lymphocyte stimulated by SIgA-F5 immune complexes (Fig. 6B-G).

Terminology: There are instances where the terminology might be confusing for readers not familiar with the subject. For example, abbreviations like "PPs", "DC", and "ICs" should be defined upon first use in both abstract and main text.

 Dendritic cell (DC) “Peyer’s patch (PP)” “M(Microfold) cell ” “subepithelial dome (SED) ”, etc.(see red part)

Results: The results section is detailed. However, it might be beneficial to provide more context or interpretation alongside the results to guide the reader.

Line 263 “These results suggested that SIgA-antigen ICs, like other antigens in gut, were mainly uptaken from PPs.”

 Discussion: The discussion section could benefit from a more in-depth comparison with existing literature. While the paper cites previous studies, a more detailed discussion on how this study advances the field would be beneficial.

Line 369  “Rapid antigen recognition and presentation produce a fast and effective immune response[24].”

Line 372 “For DC cells, as the most important antigen presentation cells, their presentation ability is also very important in mucosal immunity [26]”

Line 429 We add “There are many more questions about SIgA-antigen ICs, such as the delivery method of immune complexes, the more efficient purification method of SIgA, and the effect of large-scale application as a vaccine.”

 Comments on the Results part:

3.1: Please introduce at the beginning of the results part: what is F5-SigA, and explain the rationale of this design.

In view of this question, we have revised the title and somewhere in text to ensure that the reader can read the article more clearly.

3.2: It would be useful to mention what is green and red, not just in the Figure Legend but also in the text of Results. It could help the readers read easily.

Line 265 We add “The intestinal tissue was blue and SIgA in the complex was green. The F5 was stained with PE in red. When merging the two pictures, red encounters blue and changes to yellow, indicating that F5 and SIgA are in a protein structure. These results suggested that SIgA-antigen ICs, like other antigens in gut, were mainly uptaken from PPs.”

3.3: Please introduce what is DC2.4, is it a cell line of dendritic cells? Please also explain at the beginning of the paragraph why to use this DC2.4 model, for example, you could mention if is it commonly used to mimic in vivo dendritic cell stimulation.

Line 58 Yes, we add “DC 2.4 is a cell line of dendritic cells, which were commonly used to mimic in vivo dendritic cell stimulation.”

Figure 3: Instead of writing 1,2,3, please label the pictures with channel names, e.g., FITC, merge...Similarly, please label the use of antibodies on the left of the panels, e.g., α-CD80, α-CD86, α-CD40.

Yes, we revised the fig.3 in the text.

Please specify in the text what control was used, e.g., unstimulated DC2.4 or cells treated with PBS. This could help readers read easily.

Line 286  The sentence was add “The cells treated with PBS as control. ”

If possible, please also briefly (in one or two sentences) write the rationale and design of the experiments at the beginning of each part of the Results. These can help increase readability and guide the readers to understand the logic of the paper.

See Line 265, 282 and 306.

3.4: Are microspheres and milk spheres the same thing? Please specify.

In Fig.4 milk spheres were written wrong and corrected.

Line 180 We add “For identifying the transportation function of M cell in the model, we employed the PLGA microspheres as indicator agent for transfer experiment.”

Figure 4: Please label the X-axis with the protein name, e.g., FITC-Gp2. Remove the blue and red buttons on the right of the panels. Pay attention to the labels on the top of each panel, they are hard to read and understand, please write them in the same format.

we revised the fig.4 in the text.

Figure 4F: There seems no negative peak but there’s a negative label. Please check carefully and correct it.

we revised the fig.4F in the text.

Reviewer 2 Report

The authors demonstrated clearly that ETEC-sIgA immune complexes stimulated effectively immunocompetent cells of the intestine, using sophisticated techniques.

Major comments:

1 A thorough English language review would improve the manuscript.  It is hard for the reviewer to understand precisely even the title of the manuscript. There were so many typographical errors in the manuscript and the authors should correct them. (e.g., 'understand' in the abstract, many spelling inconsistencies, such as s-SIgA, and several sentences without precise subjects, and so on)

2. In the Author's guideline of the journal Life:

Acronyms/Abbreviations/Initialisms should be defined the first time they appear in each of three sections: the abstract; the main text; the first figure or table. When defined for the first time, the acronym/abbreviation/initialism should be added in parentheses after the written-out form.

At least, we need the written-out form of the "DC" in their title, abstract, and the first section of the manuscript for accurate understanding.

3. The authors described that IgG-type ICs can be easily prepared in the discussion section. We were wondering if the authors could provide us with some comments about the comparison between IgG-type ICs and IgA-type ICs, especially in view of the immune activation in the intestine.

We described the comments in the upper section. 

Author Response

The authors demonstrated clearly that ETEC-sIgA immune complexes stimulated effectively immunocompetent cells of the intestine, using sophisticated techniques.

Major comments:

1 A thorough English language review would improve the manuscript.  It is hard for the reviewer to understand precisely even the title of the manuscript. There were so many typographical errors in the manuscript and the authors should correct them. (e.g., 'understand' in the abstract, many spelling inconsistencies, such as s-SIgA, and several sentences without precise subjects, and so on)

The title  “Secretory IgA-ETEC F5 immune complexes promoted dendritic cells differentiation and the primed T cells proliferation in mice intestine”

 'understand' is revised to ‘understood’

All the sIgA and F5-SIgA in the text are unified as SIgA and SigA-F5(see the red part).

Many typographical errors were corrected, such as “ in vitro” “in vivo” “37°C”. All the K99 were changed to “F5”

several sentences were modified:“The inhibition test of M cell uptake way will be carried out via ingesting the immune complexes by M cells in this model. “”The SDS-PAGE was done and the proteins in gels were transferred to the PVDF membrane which was incubated by both antibodies of anti-F5 mouse serum (1:150) and goat anti-mouse IgA alpha chain (1:1000, ab97231, Abcam, Cambridge, England) for 2 h at 37°C.” “The 1% (v/v) PEST(Penicillin-Streptomycin Solution, Gibco CA, United States) were added on the upper insert side of the transwell (Corning, United States), where Caco-2 cells were seeded. ”

  1. In the Author's guideline of the journal Life:

Acronyms/Abbreviations/Initialisms should be defined the first time they appear in each of three sections: the abstract; the main text; the first figure or table. When defined for the first time, the acronym/abbreviation/initialism should be added in parentheses after the written-out form. At least, we need the written-out form of the "DC" in their title, abstract, and the first section of the manuscript for accurate understanding.

 Acronyms/Abbreviations/Initialisms were defined at the first time, such as “Dendritic cell (DC) “Peyer’s patch (PP)” “M(Microfold) cell ” “subepithelial dome (SED) ”, etc.

  1. The authors described that IgG-type ICs can be easily prepared in the discussion section. We were wondering if the authors could provide us with some comments about the comparison between IgG-type ICs and IgA-type ICs, especially in view of the immune activation in the intestine.

The sentence IgG type ICs can be easily prepared by coimmunoprecipitation or binding of monoclonal antibody to corresponding antigen. did not express the author’s intention that obtaining antiserum is usually relatively easier than preparing SIgA in mouse feces. So, this sentence is deleted. The next sentence is revised to “In our studies, the preparation of a large number of specific SIgA monoclonal antibody is very difficult, which makes the preparation of ETEC F5-SIgA immune complex becoming a technical difficulty.

Reviewer 3 Report

This is a correct study of the secretory IgA on DC and T cells in mice bowel.

While the study is technically correct, the authors should elaborate on the translational value of this research.

As a COPE endorser, I do not see why three authors share first authorship; the equal contribution is is not obvious; why are two corresponding authors necessary? This looks inappropriate in respect to COPE rules and should be changed.

just minor polishing

Author Response

This is a correct study of the secretory IgA on DC and T cells in mice bowel.

While the study is technically correct, the authors should elaborate on the translational value of this research.

Yes, Line 432 “L-Y Yu, XH, LL and GW as mentor guidance team designed the experiments. DQ, YL and XC conducted all the experiments and analyzed the experimental results, including repeated experiments.”

As a COPE endorser, I do not see why three authors share first authorship; the equal contribution is is not obvious; why are two corresponding authors necessary? This looks inappropriate in respect to COPE rules and should be changed.

Line 10  *       Correspondence: liyunyu6472@byau.edu.cn (L.Y.); Tel.: +86-4596-819-290 (L.Y.); Fax: +86-4596-819-292 (L.Y.).